# Childhood Sexual Abuse, Adult Attachment Styles, and Involvement in BDSM Practices in Adult Intimate Relationships

**DOI:** 10.3390/bs15060813

**Published:** 2025-06-13

**Authors:** Maja Selič, Vesna Jug

**Affiliations:** Faculty of Mathematics, Natural Sciences and Information Technologies, University of Primorska, 6000 Koper, Slovenia; majaseli@gmail.com

**Keywords:** BDSM practices, childhood sexual abuse, adult attachment style, adult intimate relationships

## Abstract

This study aimed to examine the role of childhood sexual abuse in attachment styles and involvement in BDSM (bondage, discipline, dominance, submission, and sadomasochism) practices in adult intimate relationships. A model was built to test the predictive value of factors for involvement in BDSM practices. This study included 318 participants. Demographic data were collected and three questionnaires were used: the Childhood Trauma Questionnaire (CTQ) identified past sexual abuse, the Adult Attachment Scale (RSQ) assessed attachment style in adulthood, and the Sadomasochism Checklist (SMCL) assessed interest in masochistic practices. The results show that childhood sexual abuse is associated with practicing and enjoying BDSM, positively with submissiveness, negatively with dominance, and positively with a composite score measuring both submissiveness and dominance. It is also linked to fearful and preoccupied attachment styles in adulthood, but not dismissive attachment. Men exhibit higher levels of submissiveness and dominance in BDSM compared to women. Older individuals are more inclined to engage in BDSM. Urban residents show higher involvement in BDSM compared to those in other environments. Homosexual or bisexual individuals in open relationships are more frequently involved in BDSM. These findings highlight the complexity of connections between past experiences, sexual preferences in BDSM, and secure attachment in intimate relationships.

## 1. Introduction

Sexuality is a multifaceted aspect of human experience that includes a wide range of identities, desires, and practices—among them being BDSM, which challenges conventional understandings of intimacy, pleasure, and power. BDSM is a form of sexual expression, with the acronym representing ([89]) (i) BD—bondage (restraint) and discipline, (ii) DS—dominance and submission, and (iii) SM—sadism, masochism, and sadomasochism. It involves consensual power exchange and/or the use of pain for sexual pleasure ([95]; [96]). Consent is central—participants agree to the behaviors involved and can withdraw consent at any time ([32]; [88]; [104]).

BDSM includes a variety of consensual practices, desires, roles, and identities ([10]; [53]). More recent and comprehensive definitions of BDSM highlight its psychological and sensory dimensions ([70]), such as the eroticization of power, emotional fulfillment, and identity exploration ([28]; [34]; [102]). Sensory elements like pain and pleasure are often enhanced by biological responses ([103]). Together, these elements form a complex framework that challenges conventional views of sexuality and intimacy ([34]; [58]). In BDSM relationships, three specific power exchanges are most common: (i) a Dom (dominant) is the person who takes control and assumes a dominant role, (ii) a Sub (submissive) relinquishes control and takes on a subordinate role, or (iii) a Switch alternates between dominant and submissive roles depending on the practices and situations ([100]).

Despite growing awareness, misconceptions persist, often rooted in early pathologizing views ([36]; [55]). Such perspectives were likely influenced by the fact that most individuals described in early literature were drawn from clinical or forensic populations, where practices were often non-consensual. These historical records reinforced the assumption that BDSM involvement reflected psychopathology, regardless of concerns for safety and consent ([29]) and influenced diagnostic classification systems. The BDSM community represents a population that has historically been subjected to unethical treatment ([54]). In response, the American Psychiatric Association introduced a distinction between non-pathological paraphilic interests and paraphilic disorders in the fifth edition of the Diagnostic and Statistical Manual of Mental Disorders (DSM-5; [6]), whereas the fourth edition (DSM-IV-TR; [5]) did not include terminology for non-pathological, atypical sexual interests.

Research suggests that interest in BDSM practices ranges from as low as 2% ([74]) to nearly 70% ([48]). This broad range may be due to differences in definitions, classification systems, and sampling biases. Nevertheless, most studies agree that BDSM-related interests are not statistically rare ([50]; [105]). Although interest in and fantasies about BDSM are relatively common, the actual engagement in BDSM practices tends to be lower, typically around 20–30% ([8]; [50]; [91]). No studies have examined the prevalence of BDSM practices in Slovenia.

Individuals who engage in BDSM consider consent a fundamental principle ([32]; [88]; [104]). The presence of informed consent among all participants differentiates BDSM from violence and/or assault ([10]). It is, of course, also important to emphasize that the same applies to all sexual activities in general. Consent is a dynamic, negotiated process involving communication, boundaries, and mutual respect ([32]; [71]). The BDSM community has adopted principles that emphasize the importance of consent, including “Safe, Sane, and Consensual” (SSC), “Risk-Aware Consensual Kink” (RACK), and “Caring, Communication, Consent, and Caution” (4Cs; [98]).

The BDSM community is notably inclusive, welcoming people of diverse sexual orientations, including heterosexual, gay, lesbian, bisexual, and transgender individuals, reflecting a pansexual trend and broadening the traditional understanding of sexual orientation ([59]). Research indicates that participation in BDSM is more common among gay, lesbian, and bisexual individuals compared to heterosexual individuals ([73]). Additionally, among gay, bisexual, and other men who have sex with men, a notable proportion participate in BDSM-related scenes, highlighting higher engagement within these sexual minority groups ([68]).

BDSM, as one of the less commonly practiced forms of sexuality among the general population, has sparked a growing body of research aimed at identifying the factors that influence individuals’ involvement in such practices. Among the most frequently examined variables are attachment to caregivers, especially insecure attachment patterns, and childhood trauma, particularly sexual abuse.

From early childhood, individuals form bonds with their parents or caregivers, whose responsiveness shapes attachment patterns that serve as prototypes for later relationships ([2]; [45]). Attachment refers to how close relationships provide a secure base for children, influencing emotional and relational development throughout life ([2]). For healthy development, caregivers must offer a stimulating and responsive environment ([99]). Attachment theory, rooted in [17]’s ([17], [18], [19]) work, is a foundational framework for understanding social and emotional development ([13]). It emphasizes the importance of caregiver sensitivity and attunement ([2]; [3]) and posits that humans seek proximity to attachment figures, especially under stress. These interactions form internal working models—mental representations of self and others—that guide emotion regulation and relationships across the lifespan ([19]; [63]).

Secure attachment develops when caregivers are consistently responsive; insecurity arises when they are unavailable or rejecting ([38]). Adult attachment is typically measured along two dimensions: anxiety (fear of abandonment) and avoidance (discomfort with closeness) ([20]; [63]). Anxious individuals use hyperactivating strategies to seek reassurance ([64]), while avoidant individuals use deactivating strategies to suppress attachment needs ([63]).

[45] ([45]) proposed that the attachment process continues throughout life and significantly impacts adult relationships. They developed three adult attachment styles: secure, anxious–ambivalent, and avoidant. Securely attached adults can form close and satisfying relationships with others, as they have a positive internal model of themselves and others. Anxiously attached adults often fear rejection or abandonment and strive to avoid these feelings. Avoidantly attached adults often avoid close relationships or are emotionally distant ([45]). [12] ([12]) further refined adult attachment theory by introducing a four-category model—secure, preoccupied, dismissing, and fearful—based on positive or negative views of self and others. These styles influence emotional regulation and relational behavior, and represent idealized prototypes that individuals may approximate to varying degrees ([25]; [49]; [78]). Figure 1 shows the four attachment patterns.

Neglect and child abuse are significant global public health concerns with long-lasting effects on victims’ psychosocial development and physical and mental health ([40]). The [101] ([101]) categorizes child abuse or maltreatment into physical, emotional, and sexual abuse, neglect or negligent treatment, and exploitation, which result in actual or potential harm to the child’s health, survival, development, or dignity in the context of a relationship of responsibility, trust, or power.

Child sexual abuse specifically involves engaging a minor in sexual activity they cannot fully understand, consent to, or are not developmentally prepared for, often to satisfy the adult’s needs ([101]). This activity between a child and an adult, who is in a relationship of responsibility, trust, or power by age or development, aims to satisfy or gratify the needs of the adult. This can include, but is not limited to ([101]), (i) inducing or coercing a child into any sexual activity, (ii) exploiting children for prostitution or other unlawful sexual practices, or (iii) exploiting children in pornography.

Research indicates that victims of childhood sexual abuse are at high risk for numerous health issues, including psychotic symptoms (especially paranoid ideas), depression, anxiety (including post-traumatic stress), obsessive-compulsive symptoms, dissociation, eating disorders, somatization ([62]), borderline personality disorder ([21]; [46]; [62]), self-image disturbances, suicidal and self-harming thoughts or behaviors ([62]), substance abuse ([62]; [77]), sexual dysfunctions, engagement in risky sexual behaviors such as unprotected sex, sex with multiple partners (this is not inherently a health issue, but it can be associated with certain health risks—particularly when protective measures and open communication are not consistently practiced), early sexual activity ([62]), prostitution ([7]; [62]), social disorders, interpersonal difficulties (including feelings of inadequacy, inferiority, or discomfort in interactions with others), hostility, anger, learning disorders, revictimization, and chronic non-cyclical pelvic pain ([62]). Importantly, a sex-positive framework encourages healthcare professionals to move beyond a risk-focused lens and adopt a more inclusive, affirming approach to sexual diversity and well-being ([67]).

Given that sexual abuse occurs within interpersonal contexts, it can disrupt the development of trust and emotional security ([22]; [41]). Survivors often report difficulties with intimacy and emotional closeness, which may impair their ability to form healthy relationships ([52]). Studies also indicate a higher prevalence of insecure attachment in adulthood among those with a history of childhood sexual abuse ([4]; [26]; [33]; [75]; [86]).

### Aim and Objectives

This study examined the role of childhood sexual abuse in shaping attachment styles and involvement in BDSM practices in adulthood. Prior research suggests that a notable proportion of individuals engaged in BDSM have experienced sexual abuse ([1]; [15]; [69]), with several studies linking such experiences to sexual masochism and masochistic fantasies ([37]; [56]; [84]; [95]). Childhood sexual abuse has also been associated with insecure attachment styles in adulthood, particularly dismissing, preoccupied, and fearful patterns ([66]), while secure attachment has been found to be negatively related to childhood victimization ([75]; [83]; [92]).

Based on previous research, the following hypotheses were formulated:Women exhibit a higher degree of submissiveness in BDSM practices than men, while men exhibit a higher degree of dominance in BDSM practices than women;Homosexual and bisexual individuals exhibit a higher degree of involvement in BDSM practices than heterosexual individuals;Childhood sexual abuse is positively associated with insecure attachment styles in adulthood;Childhood sexual abuse is positively associated with BDSM practices in adulthood, particularly with the submissive role in BDSM practices;Insecure attachment styles are positively associated with the degree of involvement in BDSM practices;Childhood sexual abuse, attachment style, gender, age, living environment, sexual orientation, and relationship status significantly predict the degree of involvement in BDSM practices.

## 2. Materials and Methods

### 2.1. Participants

The sample consisted of adult Slovenian participants. A total of 318 participants took part in the study, including 86 men (27.0%) and 230 women (72.3%), with 2 participants (0.6%) identifying their gender as other: gender non-conforming and non-binary. Table 1 presents the sample according to other demographic variables: living environment, sexual orientation, and relationship status. The average age of participants was 29.390 years, with a standard deviation of 9.377 (SD) and a range between 18 and 65 years.

### 2.2. Instruments

The battery of questionnaires included the following:A demographic questionnaire, which collected data on gender, age, living environment (rural, suburban, or urban), sexual orientation (heterosexual, homosexual, bisexual, asexual, pansexual, or other), and relationship status (single, in a relationship, married, divorced, or in an open relationship);The Relationship Scales Questionnaire (RSQ; [12]), based on a four-category model of adult attachment, which is derived from positive or negative models of self or others, resulting in four attachment categories: secure, preoccupied, dismissing, and fearful. The scale contains 30 items, and participants rate the extent to which each statement best describes their typical style in relationships on a five-point scale. Internal consistency coefficients for individual categories range from 0.87 to 0.95 ([12]).The Childhood Trauma Questionnaire (CTQ; [14]), designed to identify individuals with a history of maltreatment during childhood and adolescence. The questionnaire consists of 28 items, which participants respond to on a five-point scale. It is intended to identify five types of traumatic childhood experiences: emotional, physical, and sexual abuse, as well as emotional and physical neglect. Only the “childhood sexual abuse” subscale was used in order to maintain a focused examination of its specific associations with adult attachment styles and BDSM involvement. This targeted approach allowed for clearer interpretation of results and reduced potential confounding from other trauma types. The overall internal consistency of the questionnaire is 0.88 ([87]).For the purpose of the study, the Sadomasochism Checklist (SMCL; [94]) was adapted, originally a 24-item scale related to an individual’s interest in sadomasochistic practices. The 24 items are presented twice: once to assess interest in the dominant role and separately to assess interest in the submissive role. The SMCL offers six subscales: dominant fantasy, submissive fantasy, dominant practice, submissive practice, dominant pleasure, and submissive pleasure. Since the aim of the study was to determine participants’ involvement in BDSM practices, the subscales involving fantasies were excluded. The overall internal consistency of the questionnaire is 0.96 ([94]).

### 2.3. Procedure

After obtaining permission from the authors of the Sadomasochism Checklist (SMCL; [94]), the questionnaire was translated into Slovenian using a double-blind method. The Relationship Scales Questionnaire (RSQ; [12]) had already been translated into Slovenian for study purposes, but a back-translation had not been performed, nor had the psychometric properties been verified. Therefore, a re-translation and adaptation were conducted using the double-blind translation method. The Childhood Trauma Questionnaire (CTQ; [14]) had already been appropriately translated and was available in Slovenian. Informed consent and the battery of questionnaires were then entered into the online platform 1KA. Convenience sampling was used, as the invitation to participate in the study was posted on social media (Facebook and Instagram) and potential participants were also invited to share the invitation with acquaintances. Upon reviewing the results, it was found that there were not many responses from individuals who practice BDSM. Therefore, the invitation was also posted on a BDSM community forum (FetLife) and the Med.Over.net forum and the Society for Psychological Counseling Kameleon was asked to share the invitation on the social network Facebook. Data were collected from 2 September 2023 to 31 January 2024.

### 2.4. Statistical Analysis

Data were analyzed using Jamovi software (v.2.3.28). Descriptive statistics and Cronbach’s alpha coefficient for internal consistency of the scales were initially calculated, and the normality of variable distributions was assessed using the Shapiro–Wilk test. Spearman’s correlation coefficient was then used to examine the relationships between the variables. For group comparisons, the Mann–Whitney U test was used for comparing two groups, and the Kruskal–Wallis test was used for comparing multiple groups. Additionally, pairwise comparisons were conducted using the Dwass–Steel–Critchlow–Fligner test. Finally, linear regression analysis was conducted. Categorical predictors (gender, residential environment, sexual orientation, and relationship status) were converted into indicator (dummy) variables. The reference groups (female gender, urban residential environment, heterosexual sexual orientation, and “in a relationship” relationship status) were excluded as indicators, in accordance with procedure guidelines ([42]).

## 3. Results

In this chapter, the results of the tests and methods used to verify the hypotheses are presented. Table 2 presents the descriptive statistics and internal consistency of the Relationship Scales Questionnaire, the Childhood Trauma Questionnaire, and the Sadomasochism Checklist. The secure and preoccupied attachment styles in the Relationship Scales Questionnaire (RSQ) showed slightly lower levels of internal consistency, while the fearful and dismissing attachment styles showed higher levels of internal consistency. The Childhood Trauma Questionnaire (CTQ) and the Sadomasochism Checklist (SMCL) showed high levels of internal consistency. The results of the Shapiro–Wilk test indicated that the data are not normally distributed. Therefore, non-parametric procedures were used to test the hypotheses.

Table 3 presents correlations between variables. Several significant correlations were identified. For the purpose of correlation analysis, the dimensions of practicing and enjoying submissiveness and dominance were combined into two composite variables: practicing and enjoying BDSM practices. A negative correlation was found between secure and fearful attachment styles, as well as between secure and dismissing attachment styles. A positive correlation was observed between secure attachment and practicing submissiveness, as well as practicing and enjoying BDSM practices. A significant correlation was also found between fearful and dismissing attachment styles, as well as between fearful attachment and childhood sexual abuse. The preoccupied attachment style was negatively correlated with the dismissing attachment style, while positive correlations were observed with childhood sexual abuse, practicing submissiveness, and enjoying submissiveness. A weaker but still significant positive correlation was found with practicing and enjoying BDSM. The dismissing attachment style was positively correlated with practicing dominance and the enjoyment experienced during dominant behaviors.

Childhood sexual abuse was significantly correlated with practicing and enjoying submissiveness, dominance, and BDSM practices. Specifically, positive correlations were found with practicing and enjoying submissiveness and BDSM, while negative correlations were observed with practicing and enjoying dominance. Practicing submissiveness was significantly and positively correlated with enjoying submissiveness, practicing and enjoying dominance, and practicing and enjoying BDSM. Individuals who enjoy the submissive role showed a positive correlation with practicing and enjoying dominance, as well as with practicing and enjoying BDSM. Practicing dominance was positively correlated with enjoying dominance, as well as with practicing and enjoying BDSM practices. Practicing BDSM was positively correlated with enjoying BDSM. Age was significantly and positively correlated with secure attachment, practicing and enjoying submissiveness, practicing dominance, and practicing and enjoying BDSM. Furthermore, significant negative correlations were found between age and both the fearful and preoccupied attachment styles.

Regarding the demographic variable of gender (Table 4), only participants who identified as male or female were included in the analysis, due to the limited number of participants (*N* = 2) who identified as another gender. To ensure the objectivity of the results, the comparison was focused on male and female participants. Results indicated significant differences across all dimensions, with men reporting significantly higher levels of involvement in BDSM practices—including engagement and pleasure in submissiveness, dominance, and BDSM practices in general.

Significant differences in the expression of BDSM practices were found in relation to the residential environment (Table 5). The lowest levels of involvement in BDSM were observed among participants from rural areas. Individuals from urban areas showed the highest levels of engagement and pleasure in submissiveness and pleasure in BDSM, while those from suburban areas reported the highest levels of engagement and pleasure in dominance, as well as engagement in BDSM practices.

Significant differences were also found in the comparison based on sexual orientation with regard to engagement and pleasure in submissiveness and BDSM. However, no significant differences were observed in dominance across different sexual orientations (Table 6).

Table 7 presents the results of pairwise comparisons examining the level of involvement in BDSM practices among participants identifying as heterosexual, homosexual, and bisexual. Significant differences were found between heterosexual and homosexual participants, as well as between heterosexual and bisexual participants, in terms of engagement in submissiveness, pleasure in submissiveness, and both engagement and pleasure in BDSM practices.

Significant differences were also found in the comparison based on relationship status with regard to engagement and pleasure in submissiveness and BDSM, while no significant differences were observed in dominance based on relationship status (Table 8). It was found that participants in open relationships reported the highest average values for engaging in BDSM across all significant differences.

For the linear regression analysis (a composite variable “practicing BDSM—submission and dominance” as the criterium variable), the multiple correlation coefficient was 0.614, and the coefficient of determination was 0.377. The results of the F-test were highly significant (*p* < 0.001). As shown in Table 9, gender, bisexual and asexual sexual orientations, being in an open relationship, identifying relationship status as “other” (e.g., engaged), and older age were found to be positive predictors of practicing BDSM. In contrast, living in a rural area and being married were found to be negative predictors of practicing BDSM.

To explore potential explanatory factors for the observed group differences, mediation and moderation analyses were conducted, with childhood sexual abuse examined as a mediator or moderator between sexual orientation and the composite variable “practicing BDSM—submission and dominance”. The results (Appendix A, Table A1 and Table A2) indicated that childhood sexual abuse does not play a significant role in the relationship between sexual orientation and involvement in BDSM practices.

## 4. Discussion

The main aim of this study was to examine the role of childhood sexual abuse in attachment styles and involvement in BDSM practices within adult intimate relationships. This study explored whether childhood sexual abuse, insecure attachment style, and demographic variables (gender, age, place of residence, sexual orientation, and relationship status) are associated with the level of involvement in BDSM practices. In addition, a model was developed to test the predictive value of these factors for practicing both submission and dominance as a composite score. Based on the literature review, six hypotheses were formulated and tested using various statistical procedures and analyses.

In the first hypothesis, it was anticipated that women would exhibit a higher level of submissiveness in BDSM practices than men and that men would demonstrate a higher level of dominance compared to women. However, the findings only partially supported these expectations. Specifically, men reported a significantly higher level of both practicing and enjoying submissive roles compared to women. This result deviates from the initial assumption and contrasts with the findings of previous studies, which have generally shown that men tend to identify as dominant and women as submissive in the context of BDSM ([16]; [76]; [100]). Interestingly, some studies have reported similar patterns to those observed in the present research. For example, [65] ([65]) found that men who had experienced sexual abuse in childhood or adulthood more frequently endorsed positive beliefs about sexual submissiveness. Comparable findings were obtained in Canada by [72] ([72]), where men with a history of childhood sexual abuse reported more favorable attitudes toward sexual submission. These observations challenge traditional gender norms and sexual selection theories, which commonly associate sexual submissiveness with femininity and sexual dominance with masculinity ([51]; [97]).

Although the average differences were small, the higher levels of reported submissiveness among men in this sample mean that the first part of the hypothesis cannot be supported. This pattern suggests a divergence from traditional gender roles, which may be explained by several factors. These include shifting societal norms regarding the expression of sexuality and gender roles, increased openness to sexual exploration among men, and the influence of cultural, psychological, and identity-related variables. It is possible that the experience of adopting a submissive role may offer a sense of psychological relief or liberation to men who otherwise occupy socially dominant roles. This perspective aligns with the understanding that BDSM practices often center not only on sexual acts such as penetration and orgasm, but also on the symbolic negotiation of power, control, and vulnerability ([57]). However, framing penetration and orgasm as central overlooks the diversity of both BDSM and general sexual practices ([90]). Especially in BDSM, many techniques are non-genital and non-penetrative, making it important to move away from narrow definitions of sexual expression. The observed behavior pattern in men opens important avenues for further inquiry—particularly qualitative investigations into the subjective meanings, motivations, and psychological functions of submissiveness in men who engage in BDSM.

In contrast, the second part of the hypothesis—that men would exhibit higher levels of dominant behavior in BDSM practices—was supported by the data. Men in the sample reported a significantly higher frequency and enjoyment of dominance compared to women. This finding aligns with prior studies that have consistently indicated a greater prevalence of dominant identification among men and submissive identification among women ([16]; [76]; [97]; [100]). Dominance continues to be culturally coded as a predominantly masculine trait ([43]), while prevailing sexual norms often socialize women to associate sexuality with submissiveness and men to eroticize dominance ([51]; [79]). Moreover, fantasies involving male dominance remain among the most commonly reported sexual fantasies among women ([43]; [44]; [51]). These cultural patterns and internalized norms appear to reinforce the observed gender differences in dominant sexual roles. Therefore, the second part of the first hypothesis can be supported, as the findings are consistent with both theoretical expectations and prior empirical evidence.

In the second hypothesis, it was anticipated that homosexually and bisexually oriented individuals would exhibit a higher level of involvement in BDSM practices compared to heterosexually oriented individuals. The results revealed significant differences between heterosexual and homosexual individuals regarding the practice of and enjoyment in submission and practicing both submission and dominance as a composite score. It was observed that homosexual individuals scored higher on the significant variables (practice of and enjoyment in submission and a composite score). These findings suggest that individuals who identified as homosexual more frequently adopted and enjoyed submissive roles and more often engaged in and derived pleasure from BDSM practices, including both submission and dominance. These results are consistent with the findings of previous studies, which have indicated that the practice of BDSM is more likely among homosexual and bisexual individuals ([27]; [30]; [47]; [48]; [73]; [80]; [91]). Similarly, the results also demonstrated significant differences between heterosexual and bisexual individuals in terms of the level of practice and enjoyment in submission and the practice of both submission and dominance as a composite score. Individuals who identified as bisexual were found to practice and enjoy submission and practice both submission and dominance as a composite score more frequently than their heterosexual counterparts. This finding is again in agreement with other studies ([24]; [27]; [30]; [47]; [48]; [73]; [80]; [91]). Several studies have shown that non-heterosexual individuals report greater involvement in BDSM practices ([29]; [30]). This may reflect specific dynamics and characteristics of the LGBTQ+ minority. It is possible that LGBTQ+ individuals more easily accept and disclose dominant or submissive sexual roles and engage in BDSM practices, given that they have already faced challenges related to other stigmatized aspects of their sexuality. Therefore, the second hypothesis can be supported.

In the third hypothesis, it was assumed that childhood sexual abuse is positively associated with an insecure attachment style in adulthood. Based on the literature, an insecure attachment style was defined as dismissive, preoccupied, or fearful ([12]). The results showed a significant positive association between childhood sexual abuse and the fearful attachment style. This suggests that individuals who reported higher levels of childhood sexual abuse were more often characterized by difficulties in depending on others, low trust, and a fear of being hurt if they allowed themselves to become too close to others, which made them feel uncomfortable. The results also showed a significant positive association between childhood sexual abuse and the preoccupied attachment style. Individuals reporting higher levels of childhood sexual abuse more frequently felt discomfort when lacking close emotional relationships and expressed a strong desire for complete emotional intimacy with others. They were often concerned that others did not value them as much as they valued others and that others were unwilling to get as close as they desired.

These findings are consistent with previous research showing that secure attachment to parents and/or romantic partners is negatively associated with childhood sexual victimization in both clinical and non-clinical samples ([75]; [83]; [92]), as well as with studies confirming that childhood sexual abuse is associated with insecure attachment styles in adult relationships ([4]; [26]; [33]; [86]). No significant association was found between childhood sexual abuse and the dismissive attachment style. Therefore, the third hypothesis can be only partially supported. Based on the literature ([4]; [26]; [33]; [86]), it had been expected that a positive association would be found between childhood sexual abuse and the dismissive attachment style, as individuals may develop a negative model of others—manifesting as avoidance in relationships—and a positive model of the self, following a traumatic experience such as childhood sexual abuse. This pattern is associated with an attachment style characterized by avoidance of dependence on others and difficulties in developing trust. On the other hand, it is believed that various factors—such as the availability of social support in childhood and later life, psychological assistance, the quality of family relationships, and personality traits—can influence the development of different attachment styles.

The fourth hypothesis assumed that childhood sexual abuse Is positively associated with involvement in BDSM practices in adulthood, particularly with the submissive role in BDSM activities. The results are consistent with these expectations and confirm that childhood sexual abuse is significantly associated with involvement in and enjoyment of practicing both submission and dominance. Childhood sexual abuse also positively correlates with practicing and enjoying submissiveness and negatively correlates with practicing and enjoying dominance. This suggests that individuals who report experiencing sexual abuse in childhood more frequently report practicing and enjoying submissiveness and BDSM practices, while less frequently reporting involvement in and enjoyment of the dominant role. The findings are also in line with other studies that have examined the association between childhood sexual abuse and involvement in BDSM practices ([1]; [15]; [35]; [69]) and with studies that found childhood sexual abuse influences the adoption of submissive and masochistic roles in BDSM ([31]; [37]; [56]; [69]; [84]; [95]). Based on the obtained results, it can be concluded that individuals who were exposed to childhood sexual abuse—that is, those who experienced someone attempting to touch them in a sexual way, convincing them to watch or engage in sexual acts, threatening them in order to engage in sexual behavior, or who were sexually harassed—more frequently engage in and enjoy a position of submission and the reception of pain within BDSM practices. These may include rough sexual intercourse, stimulation through hitting, biting, pinching, choking, urination, blindfolding, genital torture, and the use of wax, whips, weights, anal plugs, or any other sex toys. Enjoyment may also involve taking commands, being disciplined, forced, verbally degraded, and assuming the role of a slave. Therefore, the fourth hypothesis can be supported.

In the fifth hypothesis, it was assumed that insecure attachment styles would be positively associated with the level of involvement in BDSM practices. Based on the literature, insecure attachment styles were defined as dismissive, preoccupied, or fearful ([12]). The dismissive attachment style was found to be significantly positively associated with the practice of and enjoyment in dominance. Individuals who more strongly endorsed the belief that others are not trustworthy and who protected themselves from anticipated rejection by avoiding close emotional connections were more likely to enjoy and engage in controlling behavior and the adoption of dominant roles. This finding is consistent with previous studies, which have shown that individuals in dominant roles tend to exhibit a more pronounced dismissive attachment style ([60]; [81]). Within the dominant role, control and power may reduce the fear of rejection. The preoccupied attachment style was positively associated with the practice of and enjoyment in submission and with the practice of both submission and dominance as a composite score. This suggests that individuals who seek approval and acceptance from others to construct their self-worth are more likely to relinquish control, adopt a submissive role, and derive pleasure from pain. This is also supported by the findings of [60] ([60]), who reported that individuals practicing submission within BDSM exhibit lower self-esteem and a stronger tendency to seek validation. From a theoretical perspective, it may be inferred that individuals engaging in submissive practices are more likely to exhibit a preoccupied attachment style, as this style involves a persistent need for approval and acceptance from others ([45]). This aligns with the submissive role in BDSM, which emphasizes complete control by the dominant and the adoption of a subordinate position. Through adherence to rules and the expression of affection toward the dominant partner, individuals in submissive roles may fulfill their internal needs. The fearful attachment style was not found to be significantly associated with any BDSM-related variable, which is consistent with several other studies that did not find a relationship between attachment styles and involvement in BDSM practices ([23]; [29]; [73]; [100]). Therefore, the fifth hypothesis can be partially supported: individuals reporting a dismissive attachment style are more likely to adopt dominant roles, while those with a preoccupied attachment style are more likely to adopt submissive roles.

In the sixth hypothesis, it was hypothesized that sexual abuse, attachment style, gender, age, living environment, sexual orientation, and relationship status would significantly predict the level of practicing both submission and dominance as a composite score. However, the results of the model indicate that male gender, bisexual and asexual orientation, being in an open relationship, identifying one’s relationship status as “other” (e.g., engaged), and older age were positive predictors of practicing both submission and dominance. These results suggest that individuals who involve in BDSM practices are more frequently older, male, bisexual or asexual, in an open relationship, or engaged. Conversely, living in a rural area and being married were found to negatively predict involvement in BDSM practices, indicating that individuals residing in rural areas or those who are married tend to participate in such practices less frequently. Contrary to expectations, childhood sexual abuse did not emerge as a significant predictor of practicing both submission and dominance, although positive correlations between these variables were observed (Table 3). This finding aligns with previous research suggesting that BDSM interests are not necessarily rooted in past trauma but may reflect a broader spectrum of sexual expression and consensual adult behavior. Based on these findings, the proposed hypothesis can be partially supported.

This study offers a significant contribution to the limited body of empirical research exploring the intersection of childhood sexual abuse, adult attachment styles, and involvement in BDSM practices within the Slovenian context. By developing a predictive model that integrates both submissive and dominant behaviors into a composite score, the research provides a nuanced understanding of how early trauma may relate to adult relational and sexual expressions. A key finding is the association between childhood sexual abuse and the development of preoccupied and fearful attachment styles—patterns known to influence adult quality of life and interpersonal functioning. These insights open important avenues for psychotherapists and mental health professionals. Tailored therapeutic approaches can be designed to support individuals with such histories, helping them cultivate secure attachment styles and, consequently, build more fulfilling and resilient relationships.

While not a direct focus of this study, the findings may contribute to a broader understanding of BDSM as a form of sexual expression that is often based on mutual consent, trust, and negotiated boundaries. The BDSM community is known to include individuals of diverse genders, sexual orientations, and personal interests. By situating BDSM within the spectrum of human sexual diversity, this perspective may support more open and respectful conversations around sexual practices and help reduce stigma associated with non-normative preferences. However, these interpretations should be viewed as contextual considerations rather than conclusions drawn directly from this study’s data.

Importantly, while a correlation was found between childhood sexual abuse and participation in BDSM—particularly in submissive roles—this study cautions against drawing causal conclusions or making generalizations. Not all individuals who engage in BDSM have experienced abuse. Many describe their practices as spiritually meaningful ([9]), avenues for personal growth ([85]), or sources of therapeutic benefit ([11]). For some, BDSM offers a structured and safe space to process past trauma ([61]) or explore complex aspects of the self ([82]). These diverse motivations highlight the potential for healing, coping, and self-development within the BDSM community.

The limitations of the conducted study should be acknowledged. Firstly, this research was based on self-reporting, which may lead to biased results. This means that participants reported on their own experiences of sexual abuse and BDSM-related sexual preferences, which can result in inaccurate information. The next limitation concerns the small sample size for certain specific sexual orientations, as the sample included a limited number of individuals with specific sexual preferences, potentially restricting the generalizability of the findings. One of the key limitations of this study is also that only two participants identified their gender as “other.” To ensure greater objectivity, they were excluded from the gender-based analysis. If there had been more participants in this category, it would have been far more informative, as it would have allowed for a meaningful comparison of variable expression across different genders. Although consent represents a fundamental and most important principle in BDSM practices, the questionnaire did not sufficiently emphasize the significance of consent, which may have led to ambiguities regarding the ethical aspects of the study and the interpretation of its results.

An important limitation of this study is also that participants did not report any prior psychopathological conditions, such as borderline personality disorder (BPD) or post-traumatic stress disorder (PTSD). In a study by [93] ([93]), it was found that women with BPD engaged in BDSM practices more frequently. Similarly, research by [39] ([39]) showed that PTSD mediated the relationship between childhood sexual abuse and masturbation motives—specifically, childhood sexual abuse was associated with higher PTSD symptoms, which in turn were linked to increased motives for masturbation related to mood improvement, relaxation, and reduced sexual arousal. It is also important to note that the exclusion of subscales related to fantasizing about BDSM practices represents a limitation of this study. Many inclusive definitions of BDSM (e.g., [48]) emphasize the element of the “mind,” which was not fully captured in this study due to the omission of those subscales. Lastly, the length of the questionnaire may have posed a barrier for some participants. Participation in a time-consuming study may discourage individuals from engaging or lead to less accurate responses, potentially resulting in distorted or incomplete information and affecting the reliability of the research findings.

In light of this study’s findings and the broader context of existing literature, several directions for future research are recommended. First, further exploration of the emotional and subjective experiences of individuals engaged in BDSM—particularly those with a history of childhood sexual abuse—could provide valuable insights. Qualitative studies may be especially useful in capturing the internal motivations, perceived therapeutic benefits, and personal meanings attributed to BDSM practices. Such research could deepen our understanding of how individuals use these practices as potential forms of self-help, coping, or self-exploration following trauma. Additionally, this study observed patterns—such as higher levels of submissiveness among male participants—that diverge from previous findings. This highlights the need for more focused investigation into male submissiveness within BDSM, a topic that remains relatively underexplored. Understanding the motivations, experiences, and social or psychological factors influencing male engagement in submissive roles could help clarify these inconsistencies and contribute to a more inclusive understanding of gender dynamics in BDSM. Future research could also examine how individuals with histories of childhood sexual abuse navigate communication, boundary-setting, and trust within BDSM relationships. Investigating the role of safe and transparent communication between partners may shed light on how these dynamics influence feelings of safety, autonomy, and self-respect. Furthermore, longitudinal studies could explore how sexual roles and preferences evolve over time, particularly in relation to earlier life experiences, including trauma. By addressing these areas, future studies can contribute to a more nuanced and empathetic understanding of BDSM practices and their psychological significance, while also informing therapeutic approaches that are sensitive to the diverse experiences of individuals who engage in them.

## 5. Conclusions

This study examined the role of childhood sexual abuse in shaping adult attachment styles and involvement in BDSM practices. Its findings indicate that while childhood sexual abuse is associated with greater involvement in submissive roles, it does not predict practicing both submission and dominance. Insecure attachment styles—particularly preoccupied and dismissive—were linked to specific BDSM preferences, suggesting possible relational dynamics such as seeking control or validation. Gender and sexual orientation emerged as significant factors as well: men reported higher levels of both submissiveness and dominance and non-heterosexual individuals showed greater involvement in BDSM. These findings challenge traditional norms and highlight the diversity of sexual expression. Importantly, this study emphasizes that BDSM is not inherently rooted in trauma. While associations with childhood sexual abuse were observed, no causal conclusions can be drawn. BDSM often reflects consensual, trust-based exploration and may serve various personal or relational functions. Despite its limitations, this study offers insights that may support a more nuanced understanding of BDSM and inform therapeutic approaches that are respectful and nonjudgmental. It also suggests directions for future qualitative research into the emotional and psychological dimensions of BDSM, particularly among individuals with histories of abuse.

## Figures and Tables

**Figure 1 behavsci-15-00813-f001:**
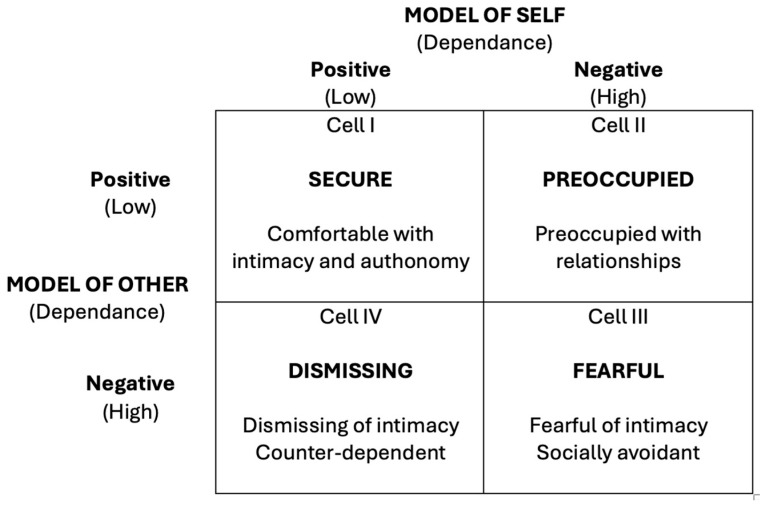
Attachment styles according to [12] ([12]).

**Table 1 behavsci-15-00813-t001:** Sample description.

Demographic Variables	*N*	%
Residential environment		
Rural	125	39.3
Suburban	66	20.8
Urban	127	39.9
Sexual orientation		
Heterosexual	234	73.6
Homosexual	14	4.4
Bisexual	52	16.4
Asexual	2	0.6
Pansexual	12	3.8
Other ^1^	4	1.3
Relationship status		
Single	101	31.8
Married	31	9.7
In a relationship	157	49.4
Open relationship	26	8.2
Divorced	3	0.9

^1^ participants identified their sexual orientation as heteroflexible, queer, or did not specify their sexual orientation.

**Table 2 behavsci-15-00813-t002:** Descriptive statistics and internal consistency of the Interpersonal Relationships Scale, Childhood Trauma Questionnaire, and Sadomasochism Checklist.

					Shapiro–Wilk		
	*M*	*SD*	Min	Max	*W*	*p*	α
Attachment style							
Secure attachment	14.063	3.03	5	23	0.974	<0.001	0.259
Fearful attachment	11.651	3.359	4	20	0.987	0.005	0.707
Preoccupied attachment	13.119	2.906	5	20	0.984	0.001	0.520
Dismissive attachment	16.425	3.968	5	25	0.967	<0.001	0.759
Childhood trauma	3.233	4.298	0	16	0.768	<0.001	0.925
Sadomasochism							
Practicing submissiveness	8.119	6.307	0	24	0.921	<0.002	0.933
Enjoyment of submissiveness	25.789	23.163	0	96	0.885	<0.001	0.953
Practicing dominance	4.987	5.394	0	24	0.807	<0.001	0.930
Enjoyment of dominance	15.629	18.825	0	88	0.776	<0.001	0.946

**Table 3 behavsci-15-00813-t003:** Correlation matrix of attachment styles, childhood sexual abuse, practicing and enjoying submissiveness and dominance, practicing and enjoying BDSM practices, and age.

		1	2	3	4	5	6	7	8	9	10	11
1	Secure attachment											
2	Fearful attachment	−0.504 ***										
3	Preoccupied attachment	−0.094	0.054									
4	Dismissive attachment	−0.231 ***	0.393 ***	−0.443 ***								
5	Childhood sexual abuse	0.018	0.176 **	0.237 ***	−0.092							
6	Practicing submissiveness	0.111 *	0.015	0.218 ***	−0.105	0.253 ***						
7	Enjoyment of submissiveness	0.104	−0.018	0.248 ***	−0.095	0.216 ***	0.874 ***					
8	Practicing dominance	0.047	−0.010	−0.010	0.116 *	−0.126 *	0.433 ***	0.372 ***				
9	Enjoyment of dominance	0.06	0.031	−0.009	0.119 *	−0.166 **	0.318 ***	0.369 ***	0.864 ***			
10	Practicing BDSM (sub. & dom.)	0.140 *	0.008	0.122 *	−0.020	0.158 **	0.878 ***	0.762 ***	0.732 ***	0.587 ***		
11	Enjoyment of BDSM (sub. & dom.)	0.143 *	0.011	0.138 *	0.003	0.140 *	0.774 ***	0.870 ***	0.639 ***	0.681 ***	0.875 ***	
12	Age	0.170 **	−0.147 **	−0.180 **	−0.018	−0.078	0.156 **	0.113**	0.227 ***	0.167	0.264 ***	0.212 ***

* *p* < 0.05. ** *p* < 0.01. *** *p* < 0.001.

**Table 4 behavsci-15-00813-t004:** Mann–Whitney U test of comparisons by gender and involvement in BDSM practices.

	*M_men*	*M_women*	*U* ^1^	*p*
Practicing submissiveness	9.860	7.409	8037.000	0.010
Enjoyment of submissiveness	32.395	23.135	8037.500	0.010
Practicing dominance	8.663	3.526	5111.500	<0.001
Enjoyment of dominance	28.081	10.678	5624.500	<0.001
Practicing BDSM (sub. & dom.)	18.523	10.935	5461.500	<0.001
Enjoyment of BDSM (sub. & dom.)	60.477	33.813	5539.500	<0.001

^1^ Mann–Whitney coefficient.

**Table 5 behavsci-15-00813-t005:** Kruskal–Wallis test of comparisons by residential environment and involvement in BDSM practices.

	*M_rural*	*M_suburban*	*M_urban*	χ^2^	*p*
Practicing submissiveness	5.920	9.152	9.748	24.881	<0.001
Enjoyment of submissiveness	18.080	26.000	33.268	21.431	<0.001
Practicing dominance	3.472	6.773	5.551	17.060	<0.001
Enjoyment of dominance	11.328	20.409	17.378	10.570	0.005
Practicing BDSM (sub. & dom.)	9.392	15.924	15.299	33.510	<0.001
Enjoyment of BDSM (sub. & dom.)	29.408	46.409	50.646	29.336	<0.001

**Table 6 behavsci-15-00813-t006:** Kruskal–Wallis test of comparisons by sexual orientation and involvement in BDSM practices.

	*M_hetero* ^1^	*M_homo* ^2^	*M_bi* ^3^	*M_a* ^4^	*M_pan* ^5^	*M_o* ^6^	χ^2^	*p*
Practicing submissiveness	6.701	14.714	11.904	0.500	11.833	11.500	44.594	<0.001
Enjoyment of submissiveness	19.825	50.571	42.346	28.000	38.667	33.000	45.900	<0.001
Practicing dominance	4.889	5.571	5.962	0.000	4.083	1.250	10.203	0.070
Enjoyment of dominance	15.064	15.714	18.865	0.000	14.667	17.000	6.676	0.246
Practic. BDSM (sub. & dom.)	11.590	20.286	17.865	0.500	15.917	12.750	38.815	<0.001
Enj. Of BDSM (sub. & dom.)	34.889	66.286	61.212	28.000	53.333	50.000	41.507	<0.001

^1^ Mean for heterosexual orientation. ^2^ Mean for homosexual orientation. ^3^ Mean for bisexual orientation. ^4^ Mean for asexual orientation. ^5^ Mean for pansexual orientation. ^6^ Mean for other sexual orientations.

**Table 7 behavsci-15-00813-t007:** Dwass–Steel–Critchlow–Fligner pairwise comparisons between heterosexuality, homosexuality, and bisexuality and the level of involvement in BDSM practices.

	*W*	*p*
Practicing submissiveness		
hetero ^1^ × homo ^2^	5.588	0.001
hetero × bi ^3^	6.713	<0.001
Enjoyment of submissiveness		
hetero × homo	5.295	0.002
hetero × bi	7.866	<0.001
Practicing dominance		
hetero × homo	0.742	0.995
hetero × bi	2.255	0.602
Enjoyment of dominance		
hetero × homo	−0.207	1.000
hetero × bi	1.529	0.889
Practicing BDSM (sub. & dom.)		
hetero × homo	5.420	0.002
hetero × bi	6.489	<0.001
Enjoyment of BDSM (sub. & dom.)		
hetero × homo	5.402	0.002
hetero × bi	7.450	<0.001

^1^ heterosexual orientation. ^2^ homosexual orientation. ^3^ bisexual orientation.

**Table 8 behavsci-15-00813-t008:** Kruskal–Wallis test of comparisons by relationship status and involvement in BDSM practices.

	*M_s* ^1^	*M_m* ^2^	*M_r* ^3^	*M_op* ^4^	*M_d* ^5^	*M_ot* ^6^	χ^2^	*p*
Practicing submissiveness	8.644	7.065	6.724	15.269	10.333	13.000	33.359	<0.001
Enjoyment of submissiveness	28.564	23.323	20.333	49.154	33.000	44.000	26.724	<0.001
Practicing dominance	4.812	4.032	4.788	7.154	7.667	19.000	10.213	0.069
Enjoyment of dominance	16.703	13.097	13.66	23.346	23.333	69.000	8.512	0.130
Practicing BDSM (sub. & dom.)	13.455	11.097	11.513	22.423	18.000	32.000	40.301	<0.001
Enjoyment of BDSM (sub. & dom.)	45.267	36.419	33.994	72.500	56.333	113.000	38.196	<0.001

^1^ Mean for single individuals. ^2^ Mean for married individuals. ^3^ Mean for those in a relationship. ^4^ Mean for those in an open relationship. ^5^ Mean for divorced individuals. ^6^ Mean for other relationship statuses.

**Table 9 behavsci-15-00813-t009:** Regression model of practicing BDSM—submission and dominance.

			95% CI				
Variable	*B*	*SE*	Lower	Upper	*t*	*p*	ß
Intercept	−3.268	6.060	−15.194	8.658	−0.539	0.590	−3.268
Attachment style							
Secure	0.166	0.185	−0.199	0.531	0.895	0.371	0.053
Fearful	0.047	0.193	−0.332	0.427	0.246	0.806	0.017
Preoccupied	0.301	0.188	−0.069	0.671	1.600	0.111	0.092
Dismissive	0.171	0.162	−0.149	0.490	1.052	0.293	0.071
Childhood trauma	0.096	0.123	−0.146	0.338	0.783	0.434	0.044
Gender							
Men	7.106	1.091	4.959	9.253	6.514	<0.001	0.333
Other	10.020	5.768	−1.331	21.371	1.737	0.083	0.084
Residential environment							
Rural	−2.689	1.052	−4.759	−0.618	−2.556	0.011	-0.139
Suburban	1.995	1.225	−0.417	4.406	1.628	0.105	0.085
Sexual orientation							
Homosexual	2.971	2.360	−1.674	7.616	1.259	0.209	0.064
Bisexual	4.597	1.268	2.102	7.092	3.627	<0.001	0.179
Asexual	−11.200	5.702	−22.421	0.021	−1.964	0.050	-0.093
Pansexual	4.679	2.432	−0.108	9.465	1.924	0.055	0.094
Other	0.569	3.955	−7.215	8.353	0.144	0.886	0.007
Relationship status							
Single	0.294	1.062	−1.797	2.385	0.277	0.782	0.014
Married	−4.273	1.672	−7.564	−0.983	−2.556	0.011	-0.134
Open relationship	6.498	1.830	2.897	10.100	3.551	<0.001	0.188
Divorced	0.381	4.620	−8.712	9.473	0.082	0.934	0.004
Other	22.303	7.927	6.702	37.904	2.813	0.005	0.132
Age	0.129	0.056	0.019	0.239	2.316	0.021	0.128

## Data Availability

The data presented in this study are not available due to privacy.

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
