# Peer review of "Childhood Sexual Abuse, Adult Attachment Styles, and Involvement in BDSM Practices in Adult Intimate Relationships"

_behavsci, 2025, doi:10.3390/bs15060813_

Round 1

Reviewer 1 Report

Comments and Suggestions for Authors

Dear Authors,
I appreciated your article on childhood sexual abuse, adult attachment styles and
involvement in BDSM practices in adult intimate relationships and the extensive work that
went behind it.
In my opinion, the study’s topic is very interesting; however, there are a few areas that
would benefit highly from some refinement. Below, I have outlined specific suggestions
for improvement:
Introduction:
- To define the dimensions mentioned at row 54 as “anxious attachment” and “avoidant
attachment” is misleading as it confuses them with the types of attachment proper.
Because of this it is better to use words such as “aspect” or similar.
- While the first part of the definition of the acronym ‘BDSM' (rows 90 and 91) could be
considered appropriate, the latter focus on the use of pain is simplistic and limiting. It
would enhance your article if you were to refer to a definition of BDSM which focuses
on three different elements: power (e.g. giving it up), sensations (e.g. sense
enhancement and deprivation) and mind (e.g. fantasies). A suggestion for further
reading that might inform this section: Pawlowski W. BDSM: The ultimate expression of
healthy sexuality. In: Taverner WJ, McKee RW, eds. Taking sides: Clashing views in
human sexuality. New York, NY: McGraw- Hill; 2009. p. 70-75.
- Additionally I would suggest for you to unify the latter paragraph (row 90-95) to the
following paragraph (row 96-110) where you mention the BDSM acronym again in order
to write a sole, abbreviated, comprehensive and concise paragraph.
- When saying that “the presence of informed consent among all participants
differentiates BDSM from violence and/or assault” I would suggest to add a small
parenthesis stating that the presence of informed consent does so for sex in general as
well. If not, a certain degree of bias might transpire when continuing to compare kinky
sex to rape.
- While anxiety and OCD can definitely have some similarities and overlapping elements,
I would suggest to refrain from defining obsessive compulsive symptoms as a part or a
subcategory of anxiety and instead to describe them separately.
- Similar to the second-to-last point, defining the sole “sex with multiple partners” as a
health issue might show some considerable sex-negative bias. For this reason I
suggest a rapid explanation on the reason/s it might be risky and possibly adding a more neutral/sex positive view of sexual practices (see for example Nimbi et al.,  2021)
Methods:
- The sections from row 220 to 227 and from 269 to 276 are almost entirely equivalent,
therefore representing somewhat of an impractical repetition. Merging the subsections
‘participants’ and ‘procedure’ might be a feasible option as well as cutting the repeated
section from the latter one.
- Following an inclusive definition of BDSM entailing the element of ‘mind’, I would
strongly suggest to cover the exclusion of the fantasy subscales of the SMCL in the
Limitations part of the study.
Results:
- The choice of excluding the two participants who identified as gender non-conforming
and non-binary should necessarily be elaborated in the Limitations part of the study.
Discussion:
- The supposed BDSM list of “rough sexual intercourse, stimulation through hitting,
biting, pinching, choking, urination, blindfolding, genital torture, and the use of wax,
whips, weights, or anal plugs” could be revised to include not only anal plugs but sex
toys in general, as any of them can be used in a BDSM scenario.
- Throughout the discussion and in regards to the six priorly stated hypotheses in the
‘Introduction’ section, it is important that you reword the results as alternative
hypotheses in research can never be confirmed. In light of this, please correct the
statements at row 419, 441, 497, 523, 551 and 568.
- The statement “BDSM practices often center not only on sexual acts such as
penetration and orgasm..” lacks overall consideration towards and acknowledgement
of not only BDSM practices but general sex practices as well. It would be extremely
beneficial to take distance from the idea that penetration and orgasm are the main/first/
central sexual acts, but particularly and especially so in a BDSM scenario where many
techniques do not require genital or penetrative acts.
- In my personal opinion the label for the composite variable is extremely confusing, as
‘practicing submission’/‘enjoying submission’ and ‘practicing dominance’/‘enjoying
dominance’ would logically be considered as BDSM practices in this context, leading
to an unclear comprehension in the text regarding the distinction between the
variables. I consequently suggest for the composite variable(s) to be identified as
‘practicing/enjoying both submission and dominance’ or akin.
- While you wrote in the ‘Abstract’ that the "results show that childhood sexual abuse is
associated with practicing and enjoying BDSM, positively with submissiveness and
negatively with dominance”, in the ‘Discussion’ it is stated that “contrary to
expectations, childhood sexual abuse did not emerge as a significant predictor of
involvement in BDSM practices”. In actuality, this ambivalence perfectly summarises
the huge inconsistencies between findings in literature and stresses how crucial it is to
approach this topic with great nuance and caution. Like in most research areas
saturated by variability and heterogeneity I would strongly suggest focusing on
understanding the unique experience of individuals, the potential which lays beyond
their resources and the healthy practices that are part of such scenarios. This can be
extremely constructive and helpful, as opposed to tunnelling on the ‘why?’ and on the
aetiology, especially because you yourself acknowledged that BDSM interests are not
necessarily rooted in past trauma (row 566) consequently highlighting the need, at the
very least, of recognition of how much stigma is at the foundation of this belief.
Conclusion:
- This last section seems somewhat more of a roundup listing instead of a connected
narration, because of this I ask that you please ensure for the conclusions to maintain
an articulate flow.
Further Comments:
- Finally, I suggest that you try to rework the whole article to be a little less discursive
and a little more concise to increase both readability and engagement.

Author Response

Dear Reviewer,

Please find our responses in the attached file.

Reviewer 2 Report

Comments and Suggestions for Authors

This manuscript addresses a highly relevant and timely topic. The integration of attachment style into the assessment framework is a valuable and innovative approach that adds depth to the understanding of the phenomena under investigation. However, several aspects of the manuscript would benefit from revision to enhance clarity, coherence, and scientific rigor.

  1. Introduction and Hypotheses:

While the topic is compelling, the introduction currently lacks a clear narrative thread that logically builds toward the study’s hypotheses. I recommend restructuring this section to establish a more coherent argument, culminating in a clear rationale for each hypothesis.

Specifically, the hypotheses would benefit from reordering—from general to specific. For instance, begin with hypotheses concerning gender differences in prevalence, followed by more specific factors such as the influence of childhood sexual abuse (CSA).

Furthermore, the inclusion of non-binary participants in the hypotheses appears insufficiently grounded in the introduction. A dedicated paragraph is needed to explain the theoretical or empirical basis for expecting differences in this group.

  1. Use of the CTQ:

It is unclear why only the CSA subscale of the Childhood Trauma Questionnaire (CTQ) was used, while other subscales were omitted. A justification for this selective use should be provided, or, alternatively, the inclusion of additional subscales considered.

  1. Role of CSA as a Covariate:

CSA may represent a key variable that helps explain group differences observed in the study. Given the elevated rates of CSA reported in LGBTQ+ populations, it would be worthwhile to explore whether CSA mediates or moderates the associations observed between gender/sexual orientation and the outcomes of interest. Including CSA as a covariate in group comparisons could clarify whether it accounts for similarities between LGBTQ+ and CSA-affected participants.

  1. Consideration of Psychopathology:

The potential impact of psychopathological conditions such as borderline personality disorder (BPD) or post-traumatic stress disorder (PTSD) should be discussed. These conditions may be relevant confounders or explanatory variables (see e.g., Warkentin et al., Borderline Personality and Emotion Processing, 2025; Gewirtz-Meydan et al., Journal of Sex & Marital Therapy, 2024).

  1. Interpretation of Findings:

The concluding statement regarding BDSM practices and societal acceptance appears overstated in light of the presented findings. While the aim to destigmatize BDSM is commendable, the current data do not provide sufficient empirical support for such broad claims. I suggest tempering this section to align more closely with the scope and limitations of the study results.

Author Response

(The authors gave the same response as above.)

Round 2

Reviewer 2 Report

Comments and Suggestions for Authors

All concerns expressed were adequately taken into account in the current revision.